# Prevalence and correlates of paediatric guideline non-adherence for initial empirical care in six low and middle-income settings: a hospital-based cross-sectional study

Riffat Ara Shawon [1,2] Donna Denno [2,3,4] Kirkby D Tickell [1,2,4]
Michael Atuhairwe,[4,5] Robert Bandsma [4,6] Ezekiel Mupere,[4,5,7]
Wieger Voskuijl [4,8,9] Emmie Mbale,[4,10] Tahmeed Ahmed [11]
Md Jobayer Chisti [11] Ali Faisal Saleem [12] Moses Ngari [13]
Abdoulaye Hama Diallo,[14] James Berkley [4,13,15] Judd Walson [1,2,3,4,16]
Arianna Rubin Means [2]

For numbered affiliations see end of article.

**Correspondence to**
Dr Riffat Ara Shawon;
riasha47@gmail.com

## ABSTRACT

**Objectives** This study evaluated the prevalence and correlates of guideline non-adherence for common childhood illnesses in low-resource settings.

**Design and setting** We used secondary cross-sectional data from eight healthcare facilities in six Asian and African countries.

**Participants** A total of 2796 children aged 2–23 months hospitalised between November 2016 and January 2019 with pneumonia, diarrhoea or severe malnutrition (SM) and without HIV infection were included in this study.

**Primary outcome measures** We identified children treated with full, partial or non-adherent initial inpatient care according to site-specific standard-of-care guidelines for pneumonia, diarrhoea and SM within the first 24 hours of admission. Correlates of guideline non-adherence were identified using generalised estimating equations.

**Results** Fully adherent care was delivered to 32% of children admitted with diarrhoea, 34% of children with pneumonia and 28% of children with SM when a strict definition of adherence was applied. Non-adherence to recommendations was most common for oxygen and antibiotics for pneumonia; fluid, zinc and antibiotics for diarrhoea; and vitamin A and zinc for SM. Non-adherence varied by site. Pneumonia guideline non-adherence was more likely among patients with severe disease (OR 1.82; 95% CI 1.38, 2.34) compared with non-severe disease. Diarrhoea guideline non-adherence was more likely among lower asset quintile groups (OR 1.16; 95% CI 1.01, 1.35), older children (OR 1.10; 95% CI 1.06, 1.13) and children presenting with wasting (OR 6.44; 95% CI 4.33, 9.57) compared with those with higher assets, younger age and not wasted.

**Conclusions** Non-adherence to paediatric guidelines was common and associated with older age, disease severity, and comorbidities, and lower household economic status. These findings highlight opportunities to improve guidelines by adding clarity to specific recommendations.

## STRENGTHS AND LIMITATIONS OF THIS STUDY

⇒ To our knowledge, this is the largest multicountry study conducted to date evaluating the prevalence of guideline non-adherence for three major causes of paediatric mortality and their subconditions: pneumonia, diarrhoea and severe malnutrition.

⇒ Strict interpretation of guidelines with the support of a clinical advisory group revealed contradictions, complexities and gaps within guidelines, including challenges in applying guidelines when children have multiple comorbidities.

⇒ This study uniquely investigated patient-level (eg, socioeconomic status), provider-level (eg, clinician-perceived risk at admission that the child may die during hospitalisation) and facility-level correlates (eg, patient arrival during weekdays vs weekends/holidays) of guideline non-adherence.

⇒ Non-adherence in this study was measured in hospitals with highly trained and monitored staff which may not be generalisable in all settings.

⇒ Observed non-adherence could be driven by purposeful deviations from guidelines, resulting from resource limitations, environmental factors (eg, local antibiotic resistance pattern) and vague or unclear guidelines. It was not possible to differentiate purposeful and accidental guideline non-adherence.

## INTRODUCTION

Globally, an estimated 14 000 under-five children die every day.[1] Leading causes of mortality in the postneonatal period include pneumonia (13%) and diarrhoea (8%), and undernutrition is an underlying cause in nearly half of child deaths under the age of 5 years.[2 3] Low-resource settings bear a disproportionate burden of child mortality, with



over 50% of these deaths occurring in sub-Saharan Africa and one-third in South Asia.[4]

WHO normative guidelines offer informed decision support for managing undernutrition, diarrhoea, pneumonia and other conditions. These guidelines are often adapted to national contexts for adoption at the country level. When applied with high fidelity, clinical guidelines can improve health outcomes of targeted populations and reduce case fatality.[5] Yet, guidelines for reducing mortality in children under five, such as the Integrated Management of Childhood Illness and other related WHO guidelines, are inconsistently applied.[6] Guideline non-adherent case management can be driven by inadequate assessment, treatment, and monitoring, local clinical culture, limitations in infrastructure, supply chain challenges, political context, differences in patient-practitioner goals, disease trajectory, patient comorbidities and clarity and applicability of recommendations. Thus, in some circumstances guideline non-adherence may be clinically justified.[7–11]

Evidence from multiple settings indicates that training alone does not affect adherence to guidelines.[12] However, there is minimal information regarding the patient, health service provider and institution-specific factors that influence fidelity to clinical guidelines in low and middle-income countries (LMICs).[13] We aim to identify the frequency of guideline non-adherence and factors influencing non-adherence during inpatient paediatric treatment for pneumonia, diarrhoea and severe malnutrition (SM) across eight facilities in six LMICs. This information will be helpful in identifying specific opportunities to support healthcare workers in their daily patient care.

## METHODS
### Study setting
This analysis used deidentified data from the Childhood Acute Illness and Nutrition (CHAIN) study, a multisite prospective cohort study that identifies risk factors for mortality in hospitalised acutely ill children and after discharge.[14] The CHAIN study took place between November 2016 and January 2019. We include children aged 2–23 months enrolled in CHAIN, hospitalised with pneumonia, diarrhoea or SM (referred to within the guidelines as severe acute malnutrition) and without HIV infection. Children with malnutrition were deliberately oversampled in CHAIN.

Urban CHAIN sites included hospitals in Karachi (Pakistan), Dhaka (Bangladesh), Kampala (Uganda), Blantyre (Malawi) and Nairobi (Kenya), and rural sites were in Banfora (Burkina Faso), Migori (Kenya) and Matlab (Bangladesh). The sites included public, private nonprofit, referral and district hospitals and often receive patients with complex or severe disease compared with lower level facilities. Further information about hospital characteristics can be found elsewhere.[15] One CHAIN site, Kilifi County Hospital, Kenya, was omitted from the analysis, as there was a potential conflict between

this analysis and ongoing internal guideline adherence auditing processes. Participating Bangladesh facilities were diarrhoea specialty centres, also admitting children with other conditions such as pneumonia and SM. In the CHAIN study, sites received a prestudy resource audit, and ongoing performance audit and feedback. Sites were also supported by trained research clinicians to ensure high standards of clinical care.[16] Thus, in this secondary analysis, we assume clinical staff were highly trained and aware of ongoing monitoring.

Key informants from each site identified institutional or national guidelines available for each condition, and if none, WHO guidelines were used for this analysis (online supplemental table 1). All sites within the same country reported using the same guidelines. Some facilities might draw from multiple guidelines to manage a child, but for the purposes of this analysis a single and most primary guideline was used as the standard of care to abstract relevant diagnostic criteria and treatment recommendations for each condition within each site. To identify children with pneumonia, diarrhoea, SM and subconditions we used CHAIN records of clinician diagnoses or, if necessary, we applied guideline diagnostic criteria (online supplemental table 1).

### Conditions and subconditions
We evaluated guideline adherence for initial treatment of pneumonia, diarrhoea and SM, the most common conditions across all sites. We also assessed adherence to subconditions, including non-severe and severe pneumonia, acute watery diarrhoea (AWD) with severe, some and no dehydration, non-severe and severe persistent diarrhoea, dysentery and SM complicated by hypoglycaemia, hypothermia, dehydration, severe anaemia, measles, micronutrient deficiency, infection and skin lesions.

We reviewed diagnoses and initial management provided within the first 24 hours of admission. However, we reviewed and counted antibiotic management up to 48 hours to allow for emergence of new clinical information.

### Defining adherence to guidelines
We formed a clinical advisory group comprising three CHAIN clinicians, two paediatricians and one general practitioner, simulating a real-world scenario of clinical decision-making with peer support. The group met four times and discussed complex clinical scenarios (eg, a child with both non-severe pneumonia and SM) to resolve ambiguity guideline interpretation when needed and to allow flexibility in situations of multimorbidities with conflicting antibiotic recommendations. For example, if a child had both non-severe pneumonia and SM, treatment with broad-spectrum injectable antibiotics (eg, ampicillin and gentamicin) recommended for SM was considered guideline adherent for non-severe pneumonia, even though pneumonia guidelines recommended an oral antibiotic (eg, amoxicillin).

Diarrhoea, pneumonia and SM guideline adherence was calculated as the proportion of children initially

treated with adherent care for all relevant subconditions within 24 hours of admission. We reported the site-specific number and proportion of children managed with full, partial or non-adherent care. Fully adherent care included providing all treatment components in guidelines (eg, recommended antibiotics, oral and intravenous fluid, therapeutic feeds). Non-adherent care indicated that a child was not provided any recommended treatments. Partial adherence indicated that some, but not all, guideline-indicated treatments were provided. We also reported adherence to specific recommendations (eg, oxygen supplementation for severe pneumonia). The analysis was agnostic to the relative importance of different recommendations.

## Statistical analysis

Descriptive statistics were applied to report the proportion of adherence to treatment for conditions (full, partial and non-adherence) and subconditions (adherence and non-adherence) within each site and overall.

The dichotomous outcome for the correlates analysis was any non-adherence to guidelines (including partial non-adherence). We explored patient-specific correlates including age, sex, socioeconomic status (composite variable standardised using asset quintiles, derived from the Demographic and Health Survey Wealth Index), preadmission care-seeking history and wasting (weight-for-height Z score <–2).[17] Other patient-specific correlates included comorbidities and presence of WHO and condition-specific danger signs (ie, obstructed breathing, respiratory distress, shock, cyanosis, convulsions, severe anaemia, severe dehydration, profuse watery diarrhoea, severe vomiting, impaired consciousness) as assessed by study clinicians.[18] Provider-specific correlates included clinician-perceived risk at admission that the child may die during hospitalisation. Facility-specific correlates included patient arrival during weekdays versus weekends/holidays according to each country's holiday and weekend conventions.

We determined the association between the prespecified correlates of interest and guideline non-adherence for each condition by using generalised estimating equations with a logit link, with clustering at the hospital level, robust SE estimates and an exchangeable correlation structure. This multivariable analysis was evaluated at a 5% alpha level.

Two independent variables had some missingness: care-seeking history (~2% missingness) and clinician risk impression on admission (~0.1% missingness). We used multiple imputation by chained equation for these variables under the assumption of data missing at random using all non-missing exposure variables in the imputation model. Rubin's rule was used to combine logistic regression outputs of 10 iterations.[19] In sensitivity analysis, we repeated all analyses with complete cases. Because the results did not substantially differ between the two approaches, we presented results from the imputed data.

Kenya's pneumonia guideline excluded children with SM from its treatment recommendations because of conflicting antibiotic recommendations in the national guidelines. Because antibiotic recommendations often conflict between SM and pneumonia recommendations, especially for non-severe pneumonia, we conducted a sensitivity analysis in which we excluded children with SM from pneumonia-specific analyses for all countries. Furthermore, all diarrhoea guidelines indicated that diarrhoea with comorbid SM requires special management, especially if the child has dehydration. Per the guidelines, SM cases were excluded when calculating diarrhoea guideline adherence for all diarrhoea subconditions. Adherence to rehydration recommendations for children with SM, diarrhoea and dehydration was assessed under the SM with dehydration subcondition.

Because SM assessment in children >6 months does not account for low birth weight or prematurity, we calculated SM guideline adherence among children <6 months versus >6 months in additional sensitivity analysis. Finally, we conducted a sensitivity analysis for conditions and subconditions without any comorbidities overall across all sites. Stata V.16 was used for all analyses.

## Role of the funding source

The funding sources had no role in study design, in the collection, analysis and interpretation of data, in the writing of or decision to submit the report for publication.

## Patient and public involvement

Patients or the public were not involved in the design, conduct, reporting or dissemination plans of our research.

## RESULTS

### Sample characteristics

Diagnosis and treatment criteria were drawn from 13 different guidelines (online supplemental tables 1 and 2) and applied to 2796 paediatric hospitalisations. A quarter of patient admissions were in Bangladesh (n=708). More than half were male (57%) and 42% were aged 12–23 months (table 1).

### Prevalence of conditions and subconditions

Approximately one-third of children were admitted with severe pneumonia and 10% with non-severe pneumonia (table 1). Over half in Kenya (57%, n=533) and Pakistan (55%, n=349) were admitted with pneumonia. About 32% of children across facilities were admitted with diarrhoea (excluding children with concurrent SM). The Bangladesh facilities, being diarrhoea specialty hospitals, had the highest proportion of children admitted with diarrhoea (81%, n=708) while Pakistan had the lowest (14%, n=349). Across facilities, 39% of children had SM, which is expected given that CHAIN purposefully overenrolled children with wasting. Approximately one-third of children were admitted with at least two of the three conditions: diarrhoea, pneumonia and SM. Online

**Table 1** Demographic characteristics and prevalence of pneumonia, diarrhoea and SM

| | Total (N=2796) | Bangladesh (n=708) | Burkina Faso (n=429) | Kenya (n=533) | Malawi (n=311) | Pakistan (n=349) | Uganda (n=466) |
|---|---|---|---|---|---|---|---|
| | N (%)* | n (%)* | n (%)* | n (%)* | n (%)* | n (%)* | n (%)* |
| **Demographics of children** | | | | | | | |
| Age at admission (months) | | | | | | | |
| 2–5 | 561 (20) | 164 (23) | 66 (15) | 122 (23) | 54 (17) | 106 (30) | 49 (11) |
| 6–11 | 1058 (37) | 313 (44) | 127 (30) | 191 (36) | 104 (33) | 122 (35) | 201 (43) |
| 12–23 | 1177 (42) | 231 (33) | 236 (55) | 220 (41) | 153 (49) | 121 (35) | 216 (46) |
| Male sex | 1588 (57) | 426 (60) | 245 (57) | 297 (56) | 172 (55) | 190 (54) | 258 (55) |
| Asset quintiles | | | | | | | |
| 1st (lowest) | 464 (17) | 58 (8) | 141 (33) | 155 (29) | 92 (30) | 1 (0.3) | 17 (4) |
| 2nd | 538 (19) | 106 (15) | 183 (43) | 71 (13) | 91 (29) | 4 (0.7) | 83 (18) |
| 3rd | 582 (21) | 140 (20) | 72 (17) | 113 (21) | 59 (19) | 30 (9) | 168 (36) |
| 4th | 596 (21) | 187 (26) | 19 (4) | 111 (21) | 34 (11) | 107 (31) | 138 (30) |
| 5th (highest) | 616 (22) | 217 (31) | 14 (3) | 83 (16) | 35 (11) | 207 (59) | 60 (13) |
| **Conditions and subconditions** | | | | | | | |
| Pneumonia | 1095 (39) | 279 (39) | 101 (24) | 306 (57) | 74 (24) | 192 (55) | 143 (31) |
| Non-severe | 296 (10) | 138 (19) | 23 (5) | 52 (10) | 27 (9) | 34 (10) | 22 (5) |
| Severe | 799 (29) | 141 (20) | 78 (18) | 254 (48) | 47 (15) | 158 (45) | 121 (26) |
| Diarrhoea† | 935 (32) | 576 (81) | 71 (16) | 103 (20) | 60 (20) | 49 (14) | 76 (16) |
| AWD with severe dehydration | 94 (3) | 31 (4) | 21 (5) | 2 (0.4) | 10 (3) | 11 (3) | 19 (4) |
| AWD with some dehydration | 130 (5) | 50 (7) | 5 (1) | 46 (9) | 5 (2) | 10 (3) | 14 (3) |
| AWD with no dehydration | 683 (23) | 494 (70) | 45 (10) | 54 (10) | 40 (13) | 16 (5) | 34 (7) |
| Persistent‡ | 14 (0.5) | | – | – | 3 (1) | 5 (1) | 6 (1) |
| Severe persistent‡ | 8 (0.3) | | – | – | 2 (1) | 4 (1) | 2 (0.4) |
| Dysentery | 6 (0.2) | 1 (0.1) | – | 1 (0.1) | – | 3 (0.9) | 1 (0.2) |
| SM§ | 1096 (39) | 269 (38) | 181 (42) | 218 (41) | 83 (27) | 118 (34) | 227 (49) |
| With hypoglycaemia | 23 (2) | 2 (0.3) | 5 (1) | 6 (1) | – | 3 (0.9) | 7 (2) |
| With hypothermia | 10 (0.9) | – | – | 4 (0.8) | 2 (0.6) | – | 4 (0.9) |
| With dehydration¶ | 174 (16) | 76 (11) | 17 (4) | 20 (4) | 43 (14) | 16 (5) | 2 (0.4) |
| With measles | 17 (2) | 2 (0.3) | – | – | – | 8 (2) | 7 (2) |
| With severe anaemia | 31 (3) | – | 27 (6) | 3 (0.6) | – | 1 (0.3) | – |
| With skin lesions | 337 (31) | 57 (8) | 35 (8) | 72 (14) | 15 (5) | 50 (14) | 109 (23) |
| Conditions with comorbidities** | | | | | | | |
| Pneumonia with other comorbidities | 905 (32) | 253 (36) | 96 (22) | 228 (43) | 54 (17) | 157 (45) | 116 (25) |
| Diarrhoea with other comorbidities | 702 (25) | 437 (62) | 70 (16) | 78 (15 | 30 (10) | 41 (12) | 46 (10) |
| SM with other comorbidities | 996 (36) | 267 (38) | 168 (39) | 206 (39) | 68 (22) | 115 (33) | 172 (37) |

Column percentages may not exactly be 100% due to rounding. Decimals are rounded to the nearest integer except when the proportion is less than 1%. No exclusions are made in this table for prevalence of conditions and subconditions unless noted otherwise. A '–' indicates there were no cases in the sample with the diagnosis. A blank cell indicates there are no specific recommendations for the given condition in the country.

*The denominator for all percentages is the total number of children treated in each site.

†Diarrhoea with dehydration in children with SM is included under SM.

‡Persistent diarrhoea involves no signs of dehydration and severe persistent diarrhoea involves signs of dehydration.

§SM subconditions are not mutually exclusive. Also, guidelines assume that all SM cases have infection, micronutrient deficiency and electrolyte imbalance and should be treated accordingly. The other subconditions listed in the table for severe acute malnutrition (SAM) may or may not be present and are not mutually exclusive.

¶Excludes SM with shock/unconsciousness based on guideline recommendations.

**Comorbidities: bronchiolitis, lower respiratory tract infection, upper respiratory tract infections, tuberculosis, otitis media, asthma, anaemia, sickle cell disease, thalassaemia, nephrotic syndrome, nephritis, other renal diseases, liver disease, ileus, cardiac disease, sepsis, malaria, soft tissue disease, urinary tract infection, measles, varicella, osteomyelitis, febrile illness, enteric fever, epilepsy, meningitis, encephalopathy, hydrocephalus, developmental delay, cerebral palsy, gastroenteritis, SM (excluding the condition originally being assessed as applicable).

AWD, acute watery diarrhoea; SM, severe malnutrition.

**Table 2** Summary of guideline adherence, by conditions and subconditions

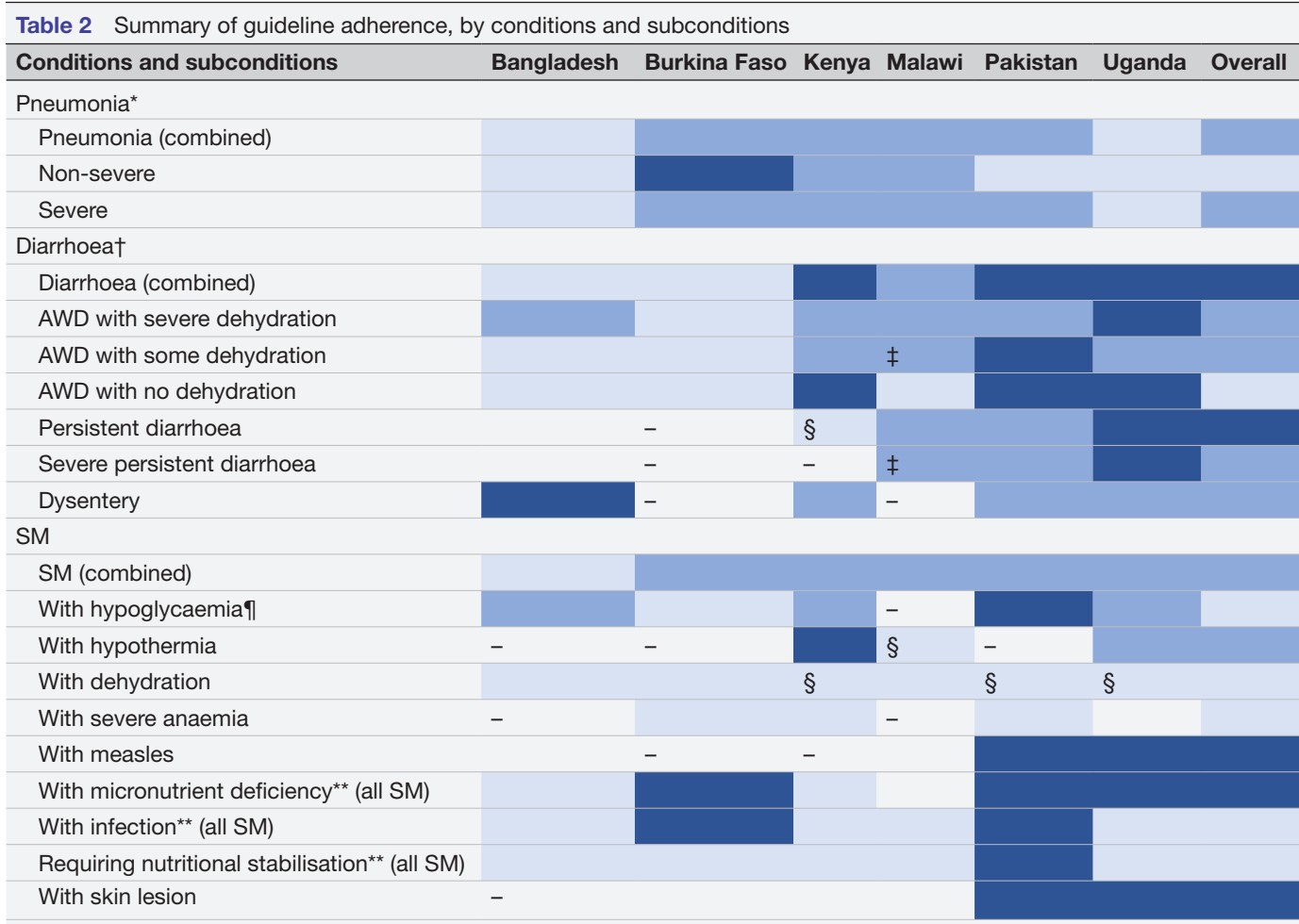

| Conditions and subconditions | Bangladesh | Burkina Faso | Kenya | Malawi | Pakistan | Uganda | Overall |
|---|---|---|---|---|---|---|---|
| **Pneumonia*** | | | | | | | |
| Pneumonia (combined) | | | | | | | |
| Non-severe | | | | | | | |
| Severe | | | | | | | |
| **Diarrhoea†** | | | | | | | |
| Diarrhoea (combined) | | | | | | | |
| AWD with severe dehydration | | | | | | | |
| AWD with some dehydration | | | | ‡ | | | |
| AWD with no dehydration | | | | | | | |
| Persistent diarrhoea | – | | § | | | | |
| Severe persistent diarrhoea | – | | – | ‡ | | | |
| Dysentery | | – | | – | | | |
| **SM** | | | | | | | |
| SM (combined) | | | | | | | |
| With hypoglycaemia¶ | | | | – | | | |
| With hypothermia | – | – | | § | – | | |
| With dehydration | | | § | | § | § | |
| With severe anaemia | – | | | – | | | |
| With measles | | – | – | | | | |
| With micronutrient deficiency** (all SM) | | | | | | | |
| With infection** (all SM) | | | | | | | |
| Requiring nutritional stabilisation** (all SM) | | | | | | | |
| With skin lesion | – | | | | | | |

A '–' indicates there were no children with the diagnosis. A blank cell indicates there are no specific recommendations for a condition in the country. Online supplemental table 4 provides detailed values and proportions for each cell.
Cell colours indicate which adherence level applied to the largest proportion of patients, by condition and site. Light blue indicates the largest proportion of patients was treated with full adherence, medium blue indicates the largest proportion of patients was treated with partial adherence and dark blue indicates the largest proportion of patients was treated with non-adherent care.
*For pneumonia, comorbid SM was excluded as per Kenyan pneumonia treatment guidelines.
†For diarrhoea, comorbid SM was excluded, per guidelines.
‡The proportion of patients treated with partially adherent and non-adherent care was equally high, and the cell is coded to partial adherence.
§The proportion of patients treated with fully adherent and non-adherent care was equally high, and the cell is coded to full adherence.
¶Hypoglycaemia treatment applies to all children with SM irrespective of serum glucose level in Uganda, Kenya, Pakistan and Bangladesh, per guidelines.
**Per guidelines, all children with SM should be treated for presumptive infection and micronutrient deficiency and require nutritional stabilisation. Micronutrients include vitamin A, zinc, folate, potassium and magnesium.
AWD, acute watery diarrhoea; SM, severe malnutrition.

supplemental table 3 provides more detail about comorbidities in the sample.

### Prevalence of guideline adherence

The guidelines used by countries have significant differences (online supplemental table 1). Prevalence of guideline adherence varied by country and guideline recommendations. Overall, 42% of children with pneumonia were treated with fully adherent care (table 2 and online supplemental table 4). The majority of the non-severe pneumonia (54%) cases were treated with full adherence, whereas only 38% of children with severe pneumonia were treated with full adherence. Lower adherence to pneumonia guidelines was driven primarily by non-provision of oxygen when recommended and prescription of different antibiotics, typically broader spectrum, from those recommended (eg, ceftriaxone rather than ampicillin and gentamicin) (table 3 and online supplemental table 5). In sensitivity analyses in which comorbid SM was excluded, we continued to observe partial adherence in most pneumonia cases (58%) (online supplemental table 6).

Across all sites, only 32% children with diarrhoea were treated with fully guideline-adherent care (online supplemental table 4). Bangladesh and Burkina Faso had the highest levels of full adherence while Kenya, Pakistan and Uganda had the lowest (table 2). The recommendation

**Table 3** Number and proportion of children treated with fully guideline-adherent care, by specific guideline recommendations

| Specific recommendations | Bangladesh<br>n (%) | Burkina Faso | Kenya | Malawi | Pakistan | Uganda | Overall |
|---|---|---|---|---|---|---|---|
| **Pneumonia*** | | | | | | | |
| Antibiotics for severe pneumonia | 107 (76) | 36 (46) | 102 (68) | 42 (89) | 0 (0) | 98 (81) | 329 (52) |
| Antibiotics for non-severe pneumonia | 101 (73) | 8 (35) | 5 (13) | 12 (44) | 17 (50) | 16 (73) | 129 (54) |
| Oxygen for severe pneumonia | | 78 (100)† | 71 (47)‡ | 6 (13)‡ | 158 (100)† | 121 (100)† | 434 (78) |
| Paracetamol for fever (severe pneumonia) | | 74 (95) | | 47 (100) | 156 (99) | 119 (98) | 396 (98) |
| Paracetamol for fever (non-severe pneumonia) | | | | 27 (100) | | 22 (100) | 49 (100) |
| Bronchodilator for wheeze (severe pneumonia) | | 67 (86) | 134 (89) | 39 (83) | 128 (81) | 109 (90) | 477 (86) |
| Bronchodilator for wheeze (non-severe pneumonia) | | | 36 (95) | 20 (74) | | 20 (91) | 76 (87) |
| **Diarrhoea§** | | | | | | | |
| Rehydration for AWD with severe dehydration | 9 (29) | 14 (67) | 2 (100) | 3 (30) | 8 (73) | 9 (47) | 45 (52) |
| Zinc for AWD with severe dehydration | 22 (71) | 13 (62) | 1 (50) | 4 (40) | 3 (27) | 3 (16) | 46 (49) |
| Rehydration for AWD with some dehydration | 49 (98) | 3 (60) | 17 (37) | 3 (60) | 1 (10) | 8 (57) | 81(80) |
| Zinc for AWD with some dehydration | | 3 (60) | 22 (48) | 1 (20) | 3 (30) | 1 (7) | 30 (28) |
| Rehydration for AWD with no dehydration | 300 (96) | 23 (51) | 15 (28) | 25 (63) | 4 (25) | 8 (24) | 375 (81) |
| No antibiotic in AWD without comorbidity | 291 (74) | 70 (99) | 82 (80) | 43 (78) | 33 (89) | 44 (66) | 651 (81) |
| Zinc for AWD with no dehydration | | 26 (58) | 14 (26) | 21 (53) | 2 (12) | 3 (9) | 66 (14) |
| Rehydration for persistent diarrhoea | | – | 2 (50) | 2 (67) | 3 (60) | 0 (0) | 7 (44) |
| Zinc for persistent diarrhoea | | – | 2 (50) | 1 (33) | 1 (20) | 0 (0) | 4 (25) |
| Multivitamins for persistent diarrhoea¶ | | – | | 0 (0) | | 0 (0) | 0 (0) |
| Microminerals** for persistent diarrhoea | | – | | 0 (0) | | | 0 (0) |
| Rehydration for severe persistent diarrhoea | | – | | 1 (50) | 3 (75) | 0 (0) | 4 (50) |
| Zinc for severe persistent diarrhoea | | – | | 0 (0) | 0 (0) | 0 (0) | 3 (30) |
| Multivitamins for severe persistent diarrhoea¶ | | – | | 0 (0) | | 0 (0) | 0 (0) |
| Microminerals** for severe persistent diarrhoea | | – | | 0 (0) | | | 0 (100) |
| Rehydration for dysentery | | – | 0 (0) | – | 2 (67) | 0 (0) | 2 (33) |
| Zinc for dysentery | | – | 0 (0) | – | 1 (33) | 0 (0) | 1 (17) |
| Antibiotics for dysentery | 0 (0) | – | 1 (100) | – | 0 (0) | 1 (100) | 2 (33) |
| **SM** | | | | | | | |
| Specific antibiotics in SM | 236 (88) | 41 (23) | 161 (74) | 62 (75) | 2 (2) | 206 (91) | 710 (65) |
| Fluid in SM with dehydration | 71 (93) | 16 (94) | 10 (50) | 35 (81) | 8 (50) | 1 (50) | 141 (33) |

Continued

**Table 3** Continued

| Specific recommendations | Bangladesh n (%) | Burkina Faso | Kenya | Malawi | Pakistan | Uganda | Overall |
|---|---|---|---|---|---|---|---|
| Blood transfusion in SM with severe anaemia | – | 21 (78) | 3 (100) | – | 1 (100) | – | 25 (81) |
| Vitamin A in SM with measles | | – | – | | 2 (25) | 1 (14) | 3 (20) |
| Micronutrients in SM | 143 (53) | 68 (38) | 114 (52) | | 22 (19) | 72 (32) | 471(43) |
| Therapeutic food in SM | 262 (97) | 153 (85) | 184 (84) | 62 (74) | 58 (49) | 147 (65) | 866 (79) |
| Zinc for skin lesion in SM | | | | | 8 (16) | 0 (0) | 8 (5) |

The denominator for all percentages is the total number of children with a specific condition. A '–' indicates there were no cases in the sample with the diagnosis. A blank cell indicates there are no relevant recommendations for a condition in the country. Online supplemental table 5 provides detailed recommendation-specific estimates of adherence levels for all cases.

Each cell indicates the number and proportion of children receiving fully adherent care. Cells coloured in dark blue indicate <65% full adherence, cells in blue indicate 65–80% full adherence and cells in light blue indicate >80% full adherence.

*For pneumonia, comorbid SM was excluded as per Kenyan pneumonia treatment guidelines.

†Oxygen is recommended for severe pneumonia with $SpO_2 < 90\%$ only.

‡Oxygen is recommended to all children with severe pneumonia.

§For diarrhoea, comorbid SM was excluded, per guidelines.

¶Vitamins include vitamin A, folate, both and/or multivitamin.

**Microminerals were an available variable in the Childhood Acute Illness and Nutrition (CHAIN) data without details of what the minerals are.

AWD, acute watery diarrhoea; SM, severe malnutrition.

commonly exhibiting least adherence across diarrhoea subconditions was provision of zinc (table 3). Persistent diarrhoea and dysentery were typically treated with partially adherent or non-adherent care across sites. Overall, 81% of children were treated will full adherence to the recommendation of withholding antibiotics for AWD without comorbidity. About 41% children with diarrhoea were admitted with comorbidities, primarily SM, for which antibiotics were indicated.

Overall, most children with SM were treated with partially adherent care (72%) (online supplemental table 4). Full adherence was observed in a minority of patients with hypothermia (20%), measles (20%) or skin lesions (5%). In contrast, most patients with SM and hypoglycaemia (47%), dehydration (81%), severe anaemia (81%) and infection (65%) were treated with fully adherent care. Nutritional stabilisation was also started in most children with SM (79%). In sensitivity analysis, we found that full adherence to overall SM guidelines was higher in children 6 months and older (29%) compared with less than 6 months (19%). Recommendations were less detailed for <6 months and simply stated that management should be similar to >6 months old, except some feeding recommendations. Full adherence was lower in children <6 months compared with 6 months and older for recommendations to treat hypoglycaemia (43% vs 54%), dehydration (79% vs 82%), presumed micronutrient deficiency (35% vs 44%) and presumed infection (61% vs 65%). Fully adherent care was higher in <6 months versus 6 months and older for treatment of skin lesion (15% vs 4%) and severe anaemia (100% vs 79%). Adherence to nutritional stabilisation recommendations did not differ between the age groups (79% fully adherent).

We conducted sensitivity analyses to describe overall guideline adherence patterns for pneumonia, diarrhoea and SM without comorbidities (online supplemental table 7) and found no appreciable difference except lower adherence for non-severe pneumonia and higher adherence for micronutrients in SM with dehydration.

### Correlates of guideline non-adherence

In multivariable analysis, children with severe pneumonia were more likely to be treated with guideline non-adherent care (OR 1.81; 95% CI 1.36, 2.41) (table 4) compared with non-severe pneumonia. For diarrhoea, children who were older (OR 1.10; 95% CI 1.06, 1.43) and wasted at admission (OR 6.79; 95% CI 4.53, 10.17) were more likely to be treated with guideline non-adherent care compared with younger or non-wasted children. Children from higher asset quintiles (OR 0.85; 95% CI 0.73, 0.99) were less likely to receive non-adherent care. For SM, none of the factors appeared to be statistically significant in this analysis.

### DISCUSSION

We examined reported paediatric treatment data from eight health facilities across a range of geographical and ecological settings in Africa and South Asia to examine adherence to standard-of-care treatment guidelines. Guidelines vary considerably across countries in level of detail and scope of included recommendations. Overall, partial adherence and non-adherence to pneumonia, diarrhoea and SM guidelines were common, although variable by country and across individual recommendations.

### Pneumonia guideline adherence

Most children with pneumonia were treated with partial adherence. Lowest adherence was noted for antibiotic

**Table 4** Correlates of pneumonia, diarrhoea and SM guideline non-adherence

| Variables | Multivariable analysis | | |
|---|---|---|---|
| | OR | 95% CI | P value |
| **Pneumonia** | | | |
| Older age | 0.98 | 0.96, 1.00 | 0.113 |
| Male sex | 0.85 | 0.66, 1.09 | 0.199 |
| Wasted at admission | 0.95 | 0.71, 1.26 | 0.706 |
| Losing weight/not gaining weight problem detected at admission | 0.80 | 0.58, 1.10 | 0.166 |
| Presence of danger sign | 1.12 | 0.81, 1.55 | 0.500 |
| Severe pneumonia* | 1.81 | 1.36, 2.41 | <0.001 |
| Previous hospitalisation | 1.08 | 0.94, 1.26 | 0.28 |
| Child likely to die† | 1.15 | 0.82, 1.63 | 0.42 |
| Asset quintile‡ | 0.99 | 0.87, 1.11 | 0.77 |
| Seeking prior treatment§ | | | |
| Did not seek treatment | 1.26 | 0.79, 2.02 | 0.332 |
| Sought treatment from shop/pharmacy | 1.02 | 0.76, 1.35 | 0.911 |
| Sought treatment from traditional healer/herbalist/homeopathist | 1.55 | 0.84, 2.88 | 0.164 |
| Presence of comorbidity | 1.08 | 0.76, 1.53 | 0.676 |
| Referred to hospital by health worker | 1.23 | 0.92, 1.66 | 0.169 |
| Admitted on a holiday | 0.87 | 0.57, 1.34 | 0.568 |
| **Diarrhoea** | | | |
| Older age | 1.10 | 1.06, 1.43 | <0.001 |
| Male sex | 1.27 | 0.91, 1.77 | 0.155 |
| Wasted at admission | 6.79 | 4.53, 10.17 | <0.001 |
| Losing weight/not gaining weight problem detected at admission¶ | 1.11 | 0.72, 1.71 | 0.649 |
| Presence of danger sign | 1.50 | 0.90, 2.50 | 0.117 |
| Previous hospitalisation | 1.01 | 0.81, 1.25 | 0.933 |
| Child likely to die† | 0.85 | 0.54, 1.35 | 0.496 |
| Asset quintile‡ | 0.85 | 0.73, 0.99 | 0.033 |
| Seeking prior treatment§ | | | |
| Did not seek treatment | 1.03 | 0.57, 1.89 | 0.900 |
| Sought treatment from shop/pharmacy | 0.91 | 0.62, 1.34 | 0.632 |
| Sought treatment from traditional healer/herbalist/homeopathist | 0.44 | 0.19, 1.06 | 0.066 |
| Presence of comorbidity | 0.87 | 0.61, 1.29 | 0.527 |
| Referred to hospital by health worker | 1.09 | 0.71, 1.67 | 0.709 |
| Admitted on a holiday | 0.86 | 0.55, 1.35 | 0.525 |
| Diarrhoea subcondition** | | | |
| AWD with severe dehydration | 1.70 | 0.96, 3.01 | 0.067 |
| AWD with some dehydration | 0.63 | 0.38, 1.06 | 0.085 |
| **SM** | | | |
| Older age | 0.98 | 0.94, 1.01 | 0.200 |
| Male sex | 1.41 | 0.93, 2.15 | 0.106 |
| Losing weight/not gaining weight problem detected at admission | 0.59 | 0.35, 1.01 | 0.056 |
| Danger sign | 0.64 | 0.35, 1.19 | 0.161 |
| Previous hospitalisation | 1.05 | 0.82, 1.33 | 0.700 |
| Child likely to die† | 0.74 | 0.44, 1.26 | 0.270 |
| Asset quintile‡ | 1.07 | 0.88, 1.31 | 0.495 |

Continued

**Table 4** Continued

| Variables | Multivariable analysis | | |
|---|---|---|---|
| | OR | 95% CI | P value |
| Presence of comorbidity | 1.34 | 0.69, 2.62 | 0.390 |
| Seeking prior treatment§ | | | |
| Did not seek treatment | 2.31 | 0.77, 7.00 | 0.137 |
| Sought treatment from shop/pharmacy | 1.39 | 0.80, 2.40 | 0.243 |
| Sought treatment from traditional healer/herbalist/homeopathist | 0.94 | 0.45, 1.99 | 0.875 |
| Referred to hospital by health worker | 0.84 | 0.53, 1.35 | 0.476 |
| Admitted on a holiday | 1.04 | 0.47, 2.31 | 0.919 |
| SM with hypoglycaemia | 0.54 | 0.18, 1.59 | 0.263 |
| SM with hypothermia | 0.72 | 0.10, 5.16 | 0.747 |
| SM with dehydration | 2.49 | 1.01, 6.13 | 0.048 |
| SM with skin lesion | 1.46 | 0.89, 2.38 | 0.132 |

There were not enough cases to run regressions with SM with measles and SM with severe anaemia.
*Non-severe pneumonia.
†Child unlikely to die.
‡Lowest quintile.
§Sought treatment from a medical facility.
¶Losing weight/not gaining weight problem detected at admission (detected by history taking) and admission mid upper arm circumferance (MUAC) group (detected by anthropometry measurement) were highly correlated. Admission MUAC group was used in the multivariable analysis as this is an objective measure.
**AWD with no dehydration.
AWD, acute watery diarrhoea; SM, severe malnutrition.

recommendations and oxygen supplementation guidance. While amoxicillin is recommended for non-severe pneumonia by all guidelines, broader spectrum antibiotics were typically prescribed in most cases. Hospitalisation is not indicated for non-severe pneumonia alone, thus it was likely that children with non-severe pneumonia had other comorbidities or were perceived by a clinician to be at risk of deterioration. A study in Sudan found only 18% of hospitalised children with pneumonia received recommended antibiotic, and that broader spectrum antimicrobials were commonly prescribed.[20] A systematic review of qualitative studies revealed that previous clinical practice, peer practices, fear of patient complications, diagnostic uncertainty, patient comorbidities, desiring a rapid cure, time pressure and cost saving influenced antibiotic prescription decisions.[21] Clinical guidelines did not emerge as a key driver of initial empirical antibiotic use in the review.[21]

Kenyan and Malawian guidelines include oxygen supplementation for all children with severe pneumonia; adherence to this recommendation was uncommon. In contrast, supplementation is recommended based on oxygen saturation levels in the other four countries' guidelines, and full adherence was observed in most children. When faced with vague or broad guidelines, healthcare workers may be required to lean on clinical judgement for decision-making on allocating limited resources such as oxygen.[22] More specific recommendations such as allocation to those at highest risk of clinical deterioration or based on oxygen saturation or specific clinical signs could help users prioritise when therapeutics are in short supply and drive higher guideline adherence.[23]

Severe pneumonia treatment recommendations contained more components and were less likely to be fully adhered to compared with non-severe pneumonia. Resource shortages, including oxygen and pulse oximetry, challenge the ability to adhere to recommendations.[24]

### Diarrhoea guideline adherence
Most children with diarrhoea were treated with guideline non-adherent care. Fluid management had the lowest adherence, including non-provision of recommended oral rehydration solution (ORS) and intravenous fluid (IVF). Vitamin A and folate for persistent diarrhoea and zinc for any form of diarrhoea were also subject to low adherence. Other studies in low-resource settings also reported low adherence to fluid and zinc recommendations.[25] This may also be driven by clinical practice culture, stockouts, lack of human resources and low provider knowledge about zinc treatment for diarrhoea.[26]

All guidelines considered in this analysis recommend that clinicians should not provide antibiotics for diarrhoea in the absence of dysentery, cholera or an indicated comorbidity. This recommendation was fully adhered to in most cases. However, for patients with dysentery, providers frequently prescribed a different antibiotic than that specified in guidelines. In contrast, a prior study in two Kenyan facilities reported frequent overprescription of initial antibiotics in non-dysentery uncomplicated diarrhoea and underprescription in dysentery.[27] Our analysis

documented and accounted for the presence of comorbidities indicative of initial antibiotic treatment, which is a novel addition to the literature. This might explain some of the overprescription of antibiotics in uncomplicated diarrhoea observed in the literature, which typically do not take comorbidities into account.

In this study, children with diarrhoea who presented with wasting were less likely to receive guideline-adherent care, compared with non-wasted children. This could result from overcautiousness of providers in dehydration management among wasted children because of the potentially narrow safety range of fluids in severe wasting.[28] Additionally, children with diarrhoea from poorer households were less likely to receive guideline-adherent care. Further investigation is necessary to determine if out-of-pocket costs present a barrier to appropriate care. Within our population of children under 2 years old, older children were also less likely to receive adherent care compared with younger children. Evidence from other diseases (malaria and lower respiratory tract infections), similarly, found older children received less guideline-adherent care.[29] Some guideline recommendations are different for younger versus older children, especially less than 6 months versus 6 months or older. For example, breast milk replaces ORS in cases requiring oral rehydration. The latter requires a hospital supply or purchase ability by the family which could lead to lower adherence in older children.

### SM guideline adherence

Most patients with SM were treated with partially adherent care in this study. A Kenyan study investigating inpatient SM guideline adherence previously reported low adherence for hypoglycaemia (15%), hypothermia (5%) and dehydration (31%) management.[30]

### Strengths and limitations

This analysis was based on a strict interpretation of guidelines. If antibiotics for pneumonia or dysentery were prescribed based on local sensitivity patterns not described in guidelines, this was considered antibiotic non-adherence. This is true particularly in Pakistan where study physicians report that they purposefully avoid first-line antibiotics due to high pathogen resistance prevalence.

We also acknowledge the potential influence of unmeasured confounders such as the clinical experience and expertise of different healthcare professionals who treated children within these hospitals in our study. Future research would benefit from systematically collecting such provider-level data to quantitatively explore these relationships and their impact on patient care outcomes.

It is possible that treatment was received without documentation as a prescription. For example, zinc or ORS could have been provided directly to caregivers without a prescription and therefore not recorded in the study database. Furthermore, it is possible that delayed documentation could have resulted in non-adherence

overestimation. These issues will be explored in future qualitative research. Additionally, data from this study were collected in the context of a cohort study in hospitals with highly trained and monitored staff and with a patient population with a disproportionately high proportion of wasted children; the frequency of guideline non-adherence observed might not be generalisable to a routine clinical care setting as a result.

Importantly, adherence in this study is not a proxy for quality of care. Non-adherence to guidelines may not always mean that care was suboptimal, rather decisions to deviate from guideline are often intentional and valid, driven by complex factors such as institutional norms, resource shortages or clinical judgement. Measuring quality of care solely based on adherence to guidelines may mask important nuances not currently captured by guidelines, or unclarity in the guidelines themselves.

### CONCLUSION

Paediatric treatment guidelines aim to consolidate evidence-based recommendations into straightforward clinical tools, relevant across cadres and settings. This study examined guideline adherence for pneumonia, diarrhoea, and SM and relevant subconditions. We found instances of both unexplained and explained guideline non-adherence. These findings highlight opportunities to improve guidelines by increasing specificity for individual recommendations. For example, clarifying oxygen supplementation recommendations for severe pneumonia, or SM management in children under 6 months of age may help improve adherence and clinical outcomes. However, this study also highlights opportunity for strategic nuance, such as providing core guideline recommendations requiring high fidelity with required resource support in addition to ensuring that recommendations are tailored at the local level (eg, use of antibiotics based on local sensitivity patterns, end-user perspectives, resources and cost considerations of treatment). This analysis also identified areas of complexity and potential confusion in guideline interpretation especially when a child has multiple conditions, underscoring the importance of addressing comorbidities in guidelines. To improve paediatric guideline adherence in low-resource settings, critical review of existing guidelines is necessary to ensure that they fully support clinicians in complex clinical scenarios.

### Author affiliations
[1]Epidemiology, University of Washington, Seattle, Washington, USA
[2]Global Health, University of Washington, Seattle, Washington, USA
[3]Pediatrics, University of Washington, Seattle, Washington, USA
[4]Childhood Acute Illness and Nutrition Network, Nairobi, Kenya
[5]Uganda-Case Western Reserve University Research Collaboration, Makerere University, Kampala, Uganda
[6]Centre for Global Child Health, Toronto, Ontario, Canada
[7]Paediatrics and Child Health, Makerere University, Kampala, Uganda
[8]Amsterdam University Medical Centres, Amsterdam, The Netherlands
[9]Kamuzu University for Health Sciences, Blantyre, Malawi
[10]Department of Pediatrics, University of Malawi, Blantyre, Malawi

¹¹International Centre for Diarrhoeal Disease Research, Dhaka, Bangladesh
¹²Aga Khan University Hospital, Karachi, Pakistan
¹³KEMRI/Wellcome Trust Research Programme, Kilifi, Kenya
¹⁴Universite Joseph Ki-Zerbo, Ouagadougou, Burkina Faso
¹⁵Centre for Tropical Medicine & Global Health, Nuffield Department of Medicine, University of Oxford, Oxford, UK
¹⁶Department of Medicine, University of Washington, Seattle, Washington, USA

**Acknowledgements** We thank the CHAIN Network for providing the data for this secondary analysis, and all CHAIN participants and hospitals and the CHAIN clinical and data management teams. We are grateful to the CHAIN sites who responded to our requests for study site information.

**Contributors** ARM, JW, DD and KDT conceptualised the study. ARM, DD, KDT and MA contributed to the methodology. ARM and KDT were responsible for data curation. ARM and RAS accessed and verified the data underlying the study. RAS did the formal analysis and prepared the original draft of the manuscript. ARM, JW, JB, DD, KDT, MA, RB, EMu, WV, EMb, MJC, TA, AFS, MN and AHD contributed to critically reviewing and editing the manuscript. RAS accepts full responsibility for the work and/or the conduct of the study, had access to the data, and controlled the decision to publish, and is responsible for the overall content as guarantor. All authors also had full access to all the data in the study and accepts responsibility to submit for publication.

**Funding** This secondary analysis of the CHAIN data was funded by the National Institutes of Health (NIH) (grant number: 5R03HD099270-02). The primary CHAIN study was supported by the Bill & Melinda Gates Foundation (grant number: OPP1131320).

**Competing interests** None declared.

**Patient and public involvement** Patients and/or the public were not involved in the design, or conduct, or reporting, or dissemination plans of this research.

**Patient consent for publication** Not applicable.

**Ethics approval** The parent CHAIN cohort study protocol was approved by the Oxford Tropical Research Ethics Committee and the institutional review boards of all partner sites. This secondary analysis was approved by the University of Washington Human Subjects Division (IRB ID: 00006878).

**Provenance and peer review** Not commissioned; externally peer reviewed.

**Data availability statement** Data are available upon reasonable request. Any reasonable requests to share data will be considered by the CHAIN consortium. Data requests should be sent to the corresponding author.

**ORCID iDs**
Riffat Ara Shawon http://orcid.org/0000-0002-3297-1982
Donna Denno http://orcid.org/0000-0002-5968-9266
Kirkby D Tickell http://orcid.org/0000-0002-4108-1236
Robert Bandsma http://orcid.org/0000-0001-6358-4750
Wieger Voskuijl http://orcid.org/0000-0001-5825-2831
Tahmeed Ahmed http://orcid.org/0000-0002-4607-7439
Md Jobayer Chisti http://orcid.org/0000-0001-9958-3071
Ali Faisal Saleem http://orcid.org/0000-0003-1804-9868
Moses Ngari http://orcid.org/0000-0001-7149-5491
James Berkley http://orcid.org/0000-0002-1236-849X
Judd Walson http://orcid.org/0000-0003-4836-720X
Arianna Rubin Means http://orcid.org/0000-0002-4087-7080

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
