## [Reviewer comments · BMJ Open]

ARTICLE DETAILS

TITLE (PROVISIONAL)	Prevalence and correlates of pediatric guideline non-adherence for initial empiric care in six low and middle-income settings: a hospital based cross sectional study
AUTHORS	Shawon , R.A; Denno, Donna; Tickell, Kirkby; Atuhairwe, Michael; Bandsma, Robert; Mupere, Ezekiel; Voskuijl, Wieger; Mbale, Emmie; Ahmed, Tahmeed; Chisti, Md.; Saleem, Ali; Ngari, Moses; DIALLO, Hama; Berkley, James; Walson, Judd; Means, Arianna

VERSION 1 – REVIEW

REVIEWER	Elias, Cecilia Universidade de Lisboa, Medicina
REVIEW RETURNED	02-Aug-2022

GENERAL COMMENTS	General The overall theme of the article is remarkably relevant. Publication of low and middle income paediatric guideline adherence is helpful to understand areas needing improvement in these settings and are an important reflection toward targeted interventions. The title requires adjustment to better represent the content of the article. The abstract is well-constructed. The background section correctly presents the context of the study and the objectives are clearly defined. The methods and analysis are well described. The results are well presented and are inkeeping with the abstract and the discussion. However, data on supplementary tables needs to be described further or corrections need to be put in place. The discussion is well written. The limitations of the study are portrayed. The conclusion responds to the proposed objectives. Supplementary tables need adjustments. Title It can be an overstatement to report these results as country results. This study is based on the analysis of 8 hospital settings across 6 countries. Furthermore, personal at these centers has been highly trained and has received feedback for performance in accordance with the study described, laying prospect that elsewhere, in these countries, guideline adherence can possibly be lower. I would advise to add the word setting to the title. Abstract - Page 6: The time period to which the study refers is not mentioned, it could enrich the abstract to include that information. Methods
---

	- Page 10: The time period to which the study refers is not mentioned, it could enrich the abstract to include that information. - Pag 13, line : “)” is missing Results Table 2 – The guidelines analysed have important differences in their degrees of completeness, for example Bangladesh pneumonia guideline solely focuses on antibiotic treatment and omits any other interventions. This means the Bangladesh hospital might present the highest guideline adherence, as per light blue colour portrayed in Table 2, however guidelines are so distinct that these reward colours might not be the best in portraying the results. Supplementary material Sup Table 4 – The top row for each condition (pneumonia, diarrhoea,..) should include all cases of the subconditions and these should add up. For instance in table 1 there are 279 cases of pneumonia in Bangladesh, however on this table there are 178 (full adherence) + 60 (none adherence). Furthermore, there are 101 non severe and 107 severe cases of pneumonia which does not add up to the 178 portrayed in the top row. Concomitantly, table 3 reports that there are 71 cases of non severe pneumonia where antibiotic was in accordance with guidelines which does not add up with the 101 reported on this table. This is recurrent throughout the table. Sup Table 5 - Same as described previously for supplementary table 4 with row sums for overall results.
--	---

REVIEWER	Cooper, Celia Womens and Childrens Hospital (WCH)
REVIEW RETURNED	30-Sep-2022

GENERAL COMMENTS	This is an interesting report on the high levels of guideline non-adherence in selected hospitals in LMIC. The study examines associations between guideline non-adherence and patient, practitioner and institutional factors and found these for all three. Unfortunately, the study was not able to question individual practitioners to identify specific factors behind their decision making. The need for additional qualitative studies to do this is noted in the limitations section. Until these are done, the reasons behind the associations remain speculative. With regard to statistical analysis, the mechanisms used to deal with missing values in the multi-variate analysis are beyond my expertise and I am asking for a statistical review.
--

REVIEWER	Hendrickson, Marissa University of Minnesota Masonic Children's Hospital, Pediatrics
REVIEW RETURNED	06-Oct-2022

GENERAL COMMENTS	The authors have undertaken a large and careful consideration of adherence to local clinical care guidelines in the care of young children presenting to hospitals in low- and middle-income settings with three common and clinically important conditions. They identified a relatively low rate of adherence to guidelines while also presenting a thoughtful consideration of potential reasons that guideline adherence may be impossible or inappropriate in certain
---

	clinical situations. Overall, I find this study to be well designed and presented. My specific questions and comments are:  1. Line 245-253 – Were these discussions conducted a priori, or in response to ambiguous situations identified in the data? If scenarios were considered a priori, was there a process for reviewing additional scenarios that were identified on review of the data? Or did the group find they were able to anticipate all reasonable challenging scenarios? Please clarify in the text. 2. Line 290-291 – “In Kenya, pneumonia guideline adherence criteria excluded children with SM presumably because of conflicting antibiotic recommendations in the national guidelines.” I find the use of “presumably” confusing here. Wasn’t the definition of adherence developed by the study team? Was excluding these children a decision that the study team made, or was it made at the local level in some way? Given the statement in lines 250-253 that the study team considered and identified an approach to this exact scenario, I am having trouble understanding why and how this was handled differently in Kenya. Please clarify. 3. Lines 294-297 – Based on the numbers in Supplemental Table 3, does this mean you excluded 41% of the diarrhea patients? Is there a reason you were not able to use the advisory group process to define a guideline-adherent approach to the care of these patients, just as you did for patients with both pneumonia and SM? Could they have been evaluated in the SM group? This is a large exclusion, which would bear mention in the discussion and/or limitations section. 4. Table 1 lists an N of 1004 for diarrhea patients, while Supplemental Table 3 lists an N of 1293. Please check the numbers and add a footnote or other explanation if there is a reason for the discrepancy. 5. Table 2 – What cutoff values for adherence levels did you use to identify a whole country as “mostly guideline adherent,” “partially guideline adherent,” or “not adherent”? Are they the same as those in Table 3? It would be helpful to include this in the table. If possible, it would improve clarity to include numbers along with the colors, as you do in Table 3. I think it would also be helpful to have a legend with a block of the color next to its definition rather than just the description “light blue,” “blue,” or “dark blue,” although this may be obviated by the journal’s formatting process. 6. Supplemental table 2 – There are three rows for diarrhea with severe dehydration with different guideline content. How is this to be interpreted? Might be helpful to add a footnote explaining this, or clarifying information in the row descriptions. 7. Supplemental table 3 – The top-line N results listed do not add up to the “total” in the far right column, nor do they match the total Ns reported elsewhere for the conditions. Is there a reason for this? Please edit the numbers to match or explain in a footnote why they differ. Minor points:  1. Table 4 – Am I correct that the rows that say “age” refer to older age? That is, that older children with diarrhea have an OR of 1.10 of receiving non-adherent care? If this is accurate, it might be helpful to replace “age” with “older age” in the table to make it clear that those ORs apply to older children, not younger ones. 2. Line 401 – “Hospitalization is not an indication...” Do you mean “Hospitalization is not indicated” or perhaps “Non-severe
--	--

	pneumonia alone is not an indication for hospitalization"? Consider rewriting for clarity. 3. Line 404 – There is a missing word after “recommended” 4. Line 444-446 – I find this confusing. “Within our population of children under 2 years old, older children were also less likely to receive adherent care compared to younger children. Evidence from other diseases (malaria and lower respiratory tract infections), in contrast, found older children received less guideline adherent care.” It says “in contrast,” but don’t the two clauses say the same thing, that older children received less-adherent care? 5. Supplemental table 8 – does “Admitted on a holiday” include weekends? Might change text to include. Thank you for the opportunity to review this interesting and well-written manuscript. I wish you all the best in this and future research endeavors.
--	---

REVIEWER	Bajpai, Ram Keele University, School of Primary, Community, and Social Care
REVIEW RETURNED	7-Oct-2022

GENERAL COMMENTS	Abstract, prevalence of adherence should be reported with the 95% confidence interval. Also, I am not sure whether the prevalence of adherence for pneumonia, diarrhoea, and severe malnutrition (SM) is a crud estimate or adjusted estimate from the model. It quite possible that crude estimate is inflated by site and country. Please provide abbreviation for OR and CI when first reported and use these acronyms. As I mentioned above, it should be clearly mentioned about how prevalence (for pneumonia, diarrhoea, and severe malnutrition) was estimated i.e., from the model or crude with a proper justification. It should also be accompanied by respective 95% confidence interval in both abstract and main manuscript. Excluding predictors based on the univariable regression with $p < 0.1$ is arbitrary and there is no theoretical justification for this cut-off. There are number of ways to such situation as well explained by Heinze et al (2018; https://doi.org/10.1002/bimj.201700067) in his article. One of the good approaches can be to information criteria such as AIC that suggests excluding any predictor with a $p \geq 0.157$ in a hierarchical fashion by first fitting the global model (i.e., full model). It is also important to report some model consistency that by leaving some predictors, model is good enough to explain variability. Need some information on whether interactions were tested or not in the multivariable model. If tested with which criteria, and if not then add some explanation. Kindly use words ‘univariable’ and ‘multivariable’ model (or regression) instead of ‘univariate’ and ‘multivariate’ as you did have one independent variable at a time. Similarly, write ‘Stata’ and not ‘STATA’ as it is not an acronym (write is as Stata 16.0 (Stata-Corp, College Station, Texas, USA)). Table 1, can author present characteristics in one column for each country in a format of ‘n (%)’ to make it visually effective? It is an optional suggestion. Although, table 2 is quite innovative but it does not pass any quantitative information (rather only showing which centre is adherent or not) and you have to read the text to know prevalence that makes it redundant. Authors should find some other way to present this data more effectively. You can simply fill data like you did in the table 3.
---

	Table 3, please add denominators that used to calculate proportion of children treated with fully guideline adherent care. Table 4, please follow suggestions as I suggested above for model building and then present them in this table. Report p-values with the three decimal points. Try to keep age and sex in all outcomes so interpretation will be much more easier as all significant predictors will be adjusted for age and sex even age or sex itself significant.
--	---

VERSION 1 – AUTHOR RESPONSE

Response to Reviewer Comments

Reviewer: 1

1. General: The overall theme of the article is remarkably relevant. Publication of low and middle income paediatric guideline adherence is helpful to understand areas needing improvement in these settings and are an important reflection toward targeted interventions. The title requires adjustment to better represent the content of the article. The abstract is well-constructed. The background section correctly presents the context of the study and the objectives are clearly defined. The methods and analysis are well described. The results are well presented and are in keeping with the abstract and the discussion. However, data on supplementary tables needs to be described further or corrections need to be put in place. The discussion is well written. The limitations of the study are portrayed. The conclusion responds to the proposed objectives. Supplementary tables need adjustments.

We appreciate your encouraging words and valuable and constructive feedback. Please see below our specific responses to your comments

2. Title: It can be an overstatement to report these results as country results. This study is based on the analysis of 8 hospital settings across 6 countries. Furthermore, personal at these centers has been highly trained and has received feedback for performance in accordance with the study described, laying prospect that elsewhere, in these countries, guideline adherence can possibly be lower. I would advise to add the word setting to the title

Thank you for your comment. We agree with your suggestion and have edited the title accordingly. The new title is *Prevalence and correlates of pediatric guideline non-adherence for initial empiric care in six low and middle-income settings*

3. Abstract: (Page 6): The time period to which the study refers is not mentioned, it could enrich the abstract to include that information.

Thank you for this helpful comment. We have now added the study dates in the abstract.

Methods: Page 10: The time period to which the study refers is not mentioned, it could enrich the abstract to include that information. Pag 13, line : “)” is missing

The study time period was added to the methods under “Study setting” section” on page 10. As mentioned above, it was also added to the abstract. The typo pointed out on Page 13 was corrected to add the missing “)”

4. Results: Table 2 – The guidelines analysed have important differences in their degrees of completeness, for example Bangladesh pneumonia guideline solely focuses on antibiotic treatment and omits any other interventions. This means the Bangladesh hospital might present the highest guideline adherent, as per light blue colour portrayed in Table 2, however guidelines are so distinct that these reward colours might not be the best in portraying the results.

Thank you for this comment. You are correct that the purpose of Table 2 is to provide information about guideline adherence in each country based upon the specific guidelines that they each individually follow. You have accurately mentioned that the guidelines have important differences across countries; the less complex the guideline the easier it is to

demonstrate full adherence. To specifically draw attention to this point to the readers, we have added the following sentences in the results section at the beginning of the descriptions for adherence levels: *There are significant differences in guidelines used in each country (Supplemental table 1) which can influence the likelihood that a country will demonstrate full adherence to all guideline recommendations (e.g. fewer guidelines recommendations may lead to higher adherence overall for a given condition).*

Supplementary material

Sup Table 4 – The top row for each condition (pneumonia, diarrhoea,..) should include all cases of the subconditions and these should add up. For instance in table 1 there are 279 cases of pneumonia in Bangladesh, however on this table there are 178 (full adherence) + 60 (none adherence).

Furthermore, there are 101 non severe and 107 severe cases of pneumonia which does not add up to the 178 portrayed in the top row. Concomitantly, table 3 reports that there are 71 cases of non severe pneumonia where antibiotic was in accordance with guidelines which does not add up with the 101 reported on this table. This is recurrent throughout the table.

Sup Table 5 - Same as described previously for supplementary table 4 with row sums for overall results.

Response: Dear reviewer, we really appreciate you catching that. There were indeed some pasting errors on the tables and supplementary tables. We have corrected those tables. However, please note that some values on the table reporting adherence levels may not add up to the total cases reported in Table 1. The table footnotes have pointed these out. For example, in Table 2, Table 3, Sup Table 4 and Sup Table 5 footnotes indicate that there were some exclusions in adherence calculation (e.g., for pneumonia, comorbid SM was excluded as per Kenyan pneumonia treatment guidelines).

Reviewer: 2

Dr. Celia Cooper, Womens and Childrens Hospital (WCH)

Comments to the Author:

This is an interesting report on the high levels of guideline non-adherence in selected hospitals in LMIC. The study examines associations between guideline non-adherence and patient, practitioner and institutional factors and found these for all three. Unfortunately, the study was not able to question individual practitioners to identify specific factors behind their decision making. The need for additional qualitative studies to do this is noted in the limitations section. Until these are done, the reasons behind the associations remain speculative.

With regard to statistical analysis, the mechanisms used to deal with missing values in the multi-variate analysis are beyond my expertise and I am asking for a statistical review.

Response: Dear Reviewer, we really appreciate your thoughtful comments. We concur with you that our qualitative study on this topic, once published, will be highly valuable in triangulating these findings and add more information from the provider perspective. We have already included a reference for the method we have used for dealing with missing data. We would be happy to clarify further and answer specific questions if raised in this review or a future one should it be required.

Reviewer: 3

Dr. Marissa Hendrickson, University of Minnesota Masonic Children's Hospital

Comments to the Author:

The authors have undertaken a large and careful consideration of adherence to local clinical care guidelines in the care of young children presenting to hospitals in low- and middle-income settings with three common and clinically important conditions. They identified a relatively low rate of adherence to guidelines while also presenting a thoughtful consideration of potential reasons that guideline adherence may be impossible or inappropriate in certain clinical situations. Overall, I find this study to be well designed and presented.

Response: Dear reviewer, we highly appreciate your kind and encouraging remarks. Please find our responses below to your specific queries.

1. Line 245-253 – Were these discussions conducted a priori, or in response to ambiguous situations identified in the data? If scenarios were considered a priori, was there a process for reviewing additional scenarios that were identified on review of the data? Or did the group find they were able to anticipate all reasonable challenging scenarios? Please clarify in the text.

Response: Thank you for this question. These discussions were carried out during this study at the time of interpreting the guidelines in order to translate these into statistical codes for the analysis. We want to emphasize that these situations did not arise from the data but from the guidelines themselves. For example, the antibiotic recommendation for non-severe pneumonia and severe malnutrition are different. If a child had both of these conditions, we sought advice from the clinical advisory group whether the child would be given two different antibiotics at the same time or just one based on the spectrum/coverage of the antibiotic. We then modified our coding in incorporate the advice so that non-adherence for antibiotic use is not overrepresented in our analysis. The following text in the manuscript reflect this idea
The group met four times and discussed complex clinical scenarios (e.g., a child with both non-severe pneumonia and SM) to resolve ambiguity guideline interpretation when needed and to allow flexibility in situations of multi-morbidities with conflicting antibiotic recommendations.

2. Line 290-291 – “In Kenya, pneumonia guideline adherence criteria excluded children with SM presumably because of conflicting antibiotic recommendations in the national guidelines.” I find the use of “presumably” confusing here. Wasn’t the definition of adherence developed by

the study team? Was excluding these children a decision that the study team made, or was it made at the local level in some way? Given the statement in lines 250-253 that the study team considered and identified an approach to this exact scenario, I am having trouble understanding why and how this was handled differently in Kenya. Please clarify.

Response: Thank you for this question. Statement in the above-mentioned lines in our submitted manuscript discussed a scenario applicable to most countries where excluding SM children was not specified in guidelines. Kenya guideline have specified this exclusion, nonetheless, without explaining the exact cause of this exclusion in their guidelines. Considering the mismatch in antibiotic recommendation in the two conditions (SM and pneumonia), in general, we ‘presumed’ that this mismatch could be the reason for the exclusion specified in Kenyan guideline, as expressed in Line 290-291 of our first submitted manuscript.

3. Lines 294-297 – Based on the numbers in Supplemental Table 3, does this mean you excluded 41% of the diarrhea patients? Is there a reason you were not able to use the advisory group process to define a guideline-adherent approach to the care of these patients, just as you did for patients with both pneumonia and SM? Could they have been evaluated in the SM group? This is a large exclusion, which would bear mention in the discussion and/or limitations section.

Response: Thank you for this question. This decision was indeed discussed with the clinical advisory group. This excluded group of children were represented in the “SM with dehydration” category. Please note that the Lines 294-297 indicate that “All diarrhea guidelines indicated that diarrhea with comorbid SM requires special management...”. In addition, our analysis only accounted for the initial management of the condition specified in the manuscript. This strategy avoided overestimation of non-adherence due to timing of the management of children with diarrhea and SM. We have now added the following sentence to the results:

Adherence to rehydration recommendations for children with diarrhea and dehydration was assessed under SM with dehydration sub-condition.

4. Table 1 lists an N of 1004 for diarrhea patients, while Supplemental Table 3 lists an N of 1293. Please check the numbers and add a footnote or other explanation if there is a reason for the discrepancy.

Response: Dear reviewer, thank you for catching this discrepancy. We have noticed pasting errors on other parts of the tables as well. We now have checked and corrected those. We sincerely apologize for this oversight.

5. Table 2 – What cutoff values for adherence levels did you use to identify a whole country as “mostly guideline adherent,” “partially guideline adherent,” or “not adherent”? Are they the same as those in Table 3? It would be helpful to include this in the table. If possible, it would improve clarity to include numbers along with the colors, as you do in Table 3. I think it would also be helpful to have a legend with a block of the color next to its definition rather than just the description “light blue,” “blue,” or “dark blue,” although this may be obviated by the journal’s formatting process.

Response: Thank you for this thoughtful suggestion. Table 2 does not use any cut off rather is just presenting whether majority of the children under the specific condition/sub-condition were treated as fully/partially/non adherent to guidelines. This is basically a simple visual presentation of the information on Supplemental Table 4. We recognize that this might not be as clear as we intended it to be. Hence, we have added the following in Table 2 footnote:

Cell colors indicate that care was mostly (e.g., the highest proportion) fully guideline adherent (light blue), partially guideline adherent (blue) or not adherent (dark blue). Please see Supplemental Table 4 for quantitative details on these adherence levels.

We look forward to the journal's formatting process to guide us more on color scheme.

6. Supplemental table 2 – There are three rows for diarrhea with severe dehydration with different guideline content. How is this to be interpreted? Might be helpful to add a footnote explaining this, or clarifying information in the row descriptions.

Response: Thank you so much for pointing this out. The titles should be diarrhea with severe dehydration, diarrhea with some dehydration and diarrhea with no dehydration. We have corrected this in our current submission.

7. Supplemental table 3 – The top-line N results listed do not add up to the “total” in the far right column, nor do they match the total Ns reported elsewhere for the conditions. Is there a reason for this? Please edit the numbers to match or explain in a footnote why they differ.

Response: Thank you for this note. There were indeed some pasting errors on the tables and supplementary tables. We have corrected those tables. However, please note that some values on the table reporting adherence levels may not add up to the total cases reported in Table 1. The table footnotes have pointed these out. For example, in Table 2, Table 3, Sup Table 4 and Sup Table 5 footnotes indicate that there were some exclusions in adherence calculation (e.g., for pneumonia, comorbid SM was excluded as per Kenyan pneumonia treatment guidelines).

Minor points:

1. Table 4 – Am I correct that the rows that say “age” refer to older age? That is, that older children with diarrhea have an OR of 1.10 of receiving non-adherent care? If this is accurate, it might be helpful to replace “age” with “older age” in the table to make it clear that those ORs apply to older children, not younger ones.

Response: Dear reviewer, thank you for pointing this out. We have replaced “age” with “older age” in the table 4.

2. Line 401 – “Hospitalization is not an indication...” Do you mean “Hospitalization is not indicated” or perhaps “Non-severe pneumonia alone is not an indication for hospitalization”? Consider rewriting for clarity.

Response: Thanks. We have rephrased it as you kindly suggested.

3. Line 404 – There is a missing word after “recommended”

Response: Thanks. We have added the missing word “antibiotic”

4. Line 444-446 – I find this confusing. “Within our population of children under 2 years old, older children were also less likely to receive adherent care compared to younger children. Evidence from other diseases (malaria and lower respiratory tract infections), in contrast, found older children received less guideline adherent care.” It says “in contrast,” but don't the two clauses say the same thing, that older children received less-adherent care?

Response: Thank you for noting this. We have replaced the phrase “in contrast,” with “also”. Apologies for this unintended oversight.

5. Supplemental table 8 – does “Admitted on a holiday” include weekends? Might change text to include.

Response: Thanks for the suggestion. Supplemental table 8 is now omitted due to change in the analytic approach in response to reviewer 4. In our new table 4, we have specified “Admitted on a holiday/weekend”.

Thank you for the opportunity to review this interesting and well-written manuscript. I wish you all the best in this and future research endeavors.

Response: We sincerely thank you for these wonderful inputs to make the manuscript better!

Reviewer: 4

Dr. Ram Bajpai, Keele University

Comments to the Author:

1. Abstract, prevalence of adherence should be reported with the 95% confidence interval. Also, I am not sure whether the prevalence of adherence for pneumonia, diarrhoea, and severe malnutrition (SM) is a crude estimate or adjusted estimate from the model. It quite possible that crude estimate is inflated by site and country. Please provide abbreviation for OR and CI when first reported and use these acronyms.

Response: Thank you for your comment. We calculated and presented the proportions. These were not derived from any modeling. Since our objective was to simply present the observed proportion of adherence for pneumonia, diarrhoea, and severe malnutrition (SM) in the countries and not estimate a population parameter from a sample, 95% confidence interval was not reported. We'd be happy to hear your further thoughts on this if needed. Additionally, we have provided the abbreviation for odds ratio (OR) and confidence interval (CI) when first reported and use these acronyms consistently throughout the manuscript.

2. As I mentioned above, it should be clearly mentioned about how prevalence (for pneumonia, diarrhoea, and severe malnutrition) was estimated i.e., from the model or crude with a proper justification. It should also be accompanied by respective 95% confidence interval in both abstract and main manuscript.

Response: Thank you for your helpful comment. We have now provided more information on how the prevalence of adherence for pneumonia, diarrhea, and severe malnutrition were calculated. As we did not adjust for site or country, we used crude estimates, which give an unambiguous statement of what the data are. Specifically, we calculated the prevalence as the proportion of adherence among cases assessed. Kindly see above comment's response for more clarification.

3. Excluding predictors based on the univariable regression with $p < 0.1$ is arbitrary and there is no theoretical justification for this cut-off. There are number of ways to such situation as well explained by Heinze et al (2018; <https://doi.org/10.1002/bimj.201700067>) in his article. One of the good approaches can be to information criteria such as AIC that suggests excluding any predictor with a $p \geq 0.157$ in a hierarchical fashion by first fitting the global model (i.e., full model). It is also important to report some model consistency that by leaving some predictors, model is good enough to explain variability.

Response: Thank you for your feedback. We acknowledge your concerns about the arbitrary nature of using a p-value threshold of 0.1 to exclude predictors in our analysis. To address this, we have revised our primary analytic approach to include all covariates in a full model, which we believe is an alternate and reasonable approach given the number of covariates of interest and the sample size.

Regarding the use of information criteria such as AIC and BIC, we consulted with a team of biostatistics consultants who advised that it is not straightforward to use these criteria with GEE since GEE does not use a full model, and these criteria are model based. While forward-selection or backward selection could be done manually, adding or removing variables depending on whether tests of their respective coefficients reach significance at some pre-specified threshold, such as 0.157 or 0.1, this approach tends to bias coefficient estimates

away from zero and understate the uncertainty we have about those estimates. Given these challenges, we believe that simply fitting all 13 covariates is the most appropriate revised approach in our analysis.

4. Need some information on whether interactions were tested or not in the multivariable model. If tested with which criteria, and if not then add some explanation.

Response: Thank you for your comment regarding interactions in the multivariable model. We did not test for any interactions in our analysis, as assessing whether and how some prespecified variable/s modifies the impact of the exposure variables on pediatric guideline adherence was not a goal of this analysis. Thank you for your valuable feedback.

5. Kindly use words 'univariable' and 'multivariable' model (or regression) instead of 'univariate' and 'multivariate' as you did have one independent variable at a time. Similarly, write 'Stata' and not 'STATA' as it is not an acronym (write is as Stata 16.0 (Stata-Corp, College Station, Texas, USA)).

Response: Thank you for your comment regarding the use of terminology in the manuscript. We have made the requested changes by using the term 'multivariable' model (or regression) instead of 'multivariate', and writing 'Stata' instead of 'STATA'. With the revision of our primary analytic approach the univariable analysis is no longer required.

6. Table 1, can author present characteristics in one column for each country in a format of 'n (%)' to make it visually effective? It is an optional suggestion.

Response: Thank you for your kind suggestion. Since this was an optional ask, we have kept our previous formatting for now.

7. Although, table 2 is quite innovative but it does not pass any quantitative information (rather only showing which centre is adherent or not) and you have to read the text to know prevalence that makes it redundant. Authors should find some other way to present this data more effectively. You can simply fill data like you did in the table 3.

Response: Thank you for your comment on our paper. We appreciate your feedback and would like to address your concern regarding Table 2. The purpose of this table is to provide a high-level visual representation of adherence to guidelines across various settings for each condition. We acknowledge that this table does not provide quantitative information, but that was not its intended purpose. To address this issue, we have provided a Supplemental Table 4, which contains detailed quantitative data on adherence to guidelines across all settings for each condition. Additionally, we have included key quantitative findings in the Results section text. We would like to argue that rather than being redundant, Table 2, Supplemental Table 4, and the Results section text are complementary and provide a comprehensive overview of our findings. We have decided to retain Table 2 as is, as it serves its intended purpose. We hope you would agree with our approach and find our paper informative.

8. Table 3, please add denominators that used to calculate proportion of children treated with fully guideline adherent care.

Response: Thank you for your comment. We truly value your feedback and want to emphasize that we made an effort to incorporate the denominators into the tables. Nevertheless, due to the presence of varying denominators for specific sub-conditions within particular countries in Table 3, a uniform denominator cannot be applied to any given column or row. Specifics regarding the number of children with particular sub-conditions in distinct settings can be found in Table 1. If you have any specific suggestions regarding how to incorporate cell-specific denominators into the table, we would greatly appreciate your input, and we're open to making further edits accordingly.

9. Table 4, please follow suggestions as I suggested above for model building and then present them in this table. Report p-values with the three decimal points. Try to keep age and sex in all outcomes so interpretation will be much more easier as all significant predictors will be adjusted for age and sex even age or sex itself significant.

Response: Thank you for your comment on our paper. We appreciate your feedback and have carefully considered your suggestions regarding Table 4. We have updated the table to include the relevant model building information, as you suggested. Additionally, we have reported the p-values with three decimal points, as per your request. Regarding your suggestion to keep age and sex in all outcomes, we have revised our primary analytical approach to adjust for all variables, which renders the issue of keeping age and sex in each analysis moot. We believe that this approach will make the interpretation of our findings much easier and more robust. We hope these changes meet your expectations and improve the clarity of our paper.

VERSION 2 – REVIEW

REVIEWER	Hendrickson, Marissa University of Minnesota Masonic Children's Hospital, Pediatrics
REVIEW RETURNED	21-Sep-2023

GENERAL COMMENTS	Thank you for the opportunity to re-review this manuscript. I am comfortable with the authors' responses to all my comments except my concern about the explanation of the colors in Table 2, expressed in comment #5. I remain uncomfortable with Table 2's presentation of colors designating mostly/partially/not adherent without including clear definitions of these findings. I think you are attempting to explain it in the footnote when you say "Cell colors indicate that care was mostly (e.g., the highest proportion) fully guideline adherent (light blue), partially guideline adherent (blue) or not adherent (dark blue). Please see Supplemental Table 4 for quantitative details on these adherence levels," but I do not find this to be adequately clear. Looking at Supplemental Table 4, it appears that you chose a color based on which of the three options (full/partial/none) had the largest number of cases in it, with full mapping to mostly, partial to partially, and none to not. Is this correct? This is one possible approach, although I question whether a site that is 48% full, 5% partial, and 47% none (or 35% full, 32% partial, and 33% none) can fairly be described as "mostly" adherent. I think an approach using numerical cutoffs, as you did in Table 3, would better fit the standard definition of "mostly." Why would you assign the colors differently between Table 2 and Table 3? In any case, given that the colors are the only data presented in this table, you should clearly explain your approach to selecting them. In addition, I have concerns about your approach when the columns were equal, as with SM with dehydration in Pakistan. I see in the notes to Supplemental Table 4 that you classified these according to the higher level of adherence, but I find that difficult to defend. If 8 patients receive fully adherent care and 8 receive non-adherent care, can that be described as "mostly" adherent? I almost missed this explanation, as it was deep in the notes of the
---

	supplemental table. It should be included in the notes for Table 2, which is where that definition was applied. I feel the numerical cutoffs used in Table 3 are likely more interesting and appropriate than reporting which level included the largest number of patients. However, if you feel strongly about sticking to your current designations, perhaps a clearer way to describe it would be: "Cell colors indicate which adherence level applied to the largest number of patients, by condition and site. Light blue indicates full adherence, medium blue indicates partial adherence, dark blue indicates non-adherence." I would place this in the subtitle, not the footnote. I also noted two small typos: Line 267 – period missing after "recommendations". Line 482 – I believe the "at" in "with wasting at were" is erroneous.
--	--

REVIEWER	Elias, Cecilia Universidade de Lisboa, Medicina
REVIEW RETURNED	22-Sep-2023

GENERAL COMMENTS	Manuscript greatly improved. Ready for publication.
--

REVIEWER	Bajpai, Ram Keele University, School of Primary, Community, and Social Care
REVIEW RETURNED	02-Dec-2023

GENERAL COMMENTS	Did authors account for the between county heterogeneity in the regression analysis, and also check for any potential interaction between the covariates when running the logistic regression? Did authors try to quantify any rural-urban, and type of health infrastructure effect in the analyses as these will have potential impact on the adherence level? I think Table 1 needs a better presentation. Number and respective percent value can be presented as 'n (%)' in the same column for each country. The total column should be the first column, and authors should also avoid the vertical separating lines. Table 2 is quite innovative showing different levels of adherence using different shades of blue colour. However, it does not provide the range/cut-off for each level of adherence either in the title or in the table footnote. I think Table 3 better suits in the supplementary as Table 2 already provided the summary of the adherence level and its results can be accompanied with the Table 2 description. Why did authors chose reporting only univariate analysis in the Table 4 although its description suggests multivariable regression analysis? Please check and rectify as appropriate. Limitation section should also be addressing the issue of residual confounding in the analysis.
---

VERSION 2 – AUTHOR RESPONSE

Reviewer: 1

Dr. Marissa Hendrickson, University of Minnesota Masonic Children's Hospital

Comments to the Author:

Thank you for the opportunity to re-review this manuscript. I am comfortable with the authors' responses to all my comments except my concern about the explanation of the colors in Table 2, expressed in comment #5.

I remain uncomfortable with Table 2's presentation of colors designating mostly/partially/not adherent without including clear definitions of these findings. I think you are attempting to explain it in the footnote when you say "Cell colors indicate that care was mostly (e.g., the highest proportion) fully guideline adherent (light blue), partially guideline adherent (blue) or not adherent (dark blue). Please see Supplemental Table 4 for quantitative details on these adherence levels," but I do not find this to be adequately clear.

Looking at Supplemental Table 4, it appears that you chose a color based on which of the three options (full/partial/none) had the largest number of cases in it, with full mapping to mostly, partial to partially, and none to not. Is this correct? This is one possible approach, although I question whether a site that is 48% full, 5% partial, and 47% none (or 35% full, 32% partial, and 33% none) can fairly be described as "mostly" adherent. I think an approach using numerical cutoffs, as you did in Table 3, would better fit the standard definition of "mostly." Why would you assign the colors differently between Table 2 and Table 3? In any case, given that the colors are the only data presented in this table, you should clearly explain your approach to selecting them.

In addition, I have concerns about your approach when the columns were equal, as with SM with dehydration in Pakistan. I see in the notes to Supplemental Table 4 that you classified these according to the higher level of adherence, but I find that difficult to defend. If 8 patients receive fully adherent care and 8 receive non-adherent care, can that be described as "mostly" adherent? I almost missed this explanation, as it was deep in the notes of the supplemental table. It should be included in the notes for Table 2, which is where that definition was applied.

I feel the numerical cutoffs used in Table 3 are likely more interesting and appropriate than reporting which level included the largest number of patients. However, if you feel strongly about sticking to your current designations, perhaps a clearer way to describe it would be: "Cell colors indicate which adherence level applied to the largest number of patients, by condition and site. Light blue indicates full adherence, medium blue indicates partial adherence, dark blue indicates non-adherence." I would place this in the subtitle, not the footnote.

Response:

Thank you for your very thoughtful response and ideas. We are grateful for your continued feedback, which has significantly contributed to the refinement of our manuscript through all stages of revision. Your concern about the potential for misinterpretation arising from Table 2 has been carefully considered. We also thank you for offering alternatives to rectify this issue.

In line with your second suggested solution, we have edited the Table 2 subtitle that includes the description "Cell colors indicate which adherence level applied to the largest proportion of patients, by condition and site. Light blue indicates a majority of patients were treated with full adherence, medium blue indicates a majority of patients were treated with partial adherence, and dark blue indicates a

majority of patients were treated with non-adherent care". Our rationale to choose this option was to provide a visual summary of Supplemental Table 4 that is easier to digest, which is particularly important because partial adherence was common for many sub-conditions.

To clarify instances where two adherence levels shared the same proportion, we have introduced superscripts in the relevant cells of Table 2 and provided explanatory notes in the corresponding table footnotes.

Table 3 is structurally different as partial adherence is not possible for any given row, each being a specific guideline recommendation. We appreciate the opportunity to enhance the manuscript with these revisions and trust they meet your approval.

I have concerns about your approach when the columns were equal, as with SM with dehydration in Pakistan. I see in the notes to Supplemental Table 4 that you classified these according to the higher level of adherence, but I find that difficult to defend. If 8 patients receive fully adherent care and 8 receive non-adherent care, can that be described as "mostly" adherent?

I also noted two small typos:

Line 267 – period missing after "recommendations".

Line 482 – I believe the "at" in "with wasting at were" is erroneous.

Response:

Thank you for your careful review. We did not find a period missing after the word "recommendations" on line 267 or elsewhere in the document. However, we have done a full, final review of the text, including removing the superfluous "at" on line 482.

Reviewer: 2

Dr. Cecilia Elias, Universidade de Lisboa

Comments to the Author:

Manuscript greatly improved.

Ready for publication.

Response:

We sincerely appreciate your positive feedback and are pleased to hear that the manuscript meets your approval for publication. We are grateful for your contribution to the improvement of our manuscript.

Reviewer: 3

Dr. Ram Bajpai, Keele University

Comments to the Author:

Did authors account for the between county heterogeneity in the regression analysis, and also check for any potential interaction between the covariates when running the logistic regression?

Response:

Dear reviewer, thank you for your insightful question. As described in the methods, we have 8 hospital

sites within 6 countries. Each of these hospitals are either rural or urban in location. We acknowledge that there would be differences and similarities in each hospital in terms of resources and practices etc. based on their location.

We have performed Generalized Estimating Equations (GEE) using clustering at the hospital-level with robust standard error estimates. This method ensured that our estimates are robust to the clustering of data by location. Additionally, this approach implicitly adjusted for some between-country variability by allowing for different baseline responses across the strata.

To further clarify, our research question focused on a population average (i.e., marginal estimates) approach to estimate the average effect of correlates across all country's hospitals within the study rather than focusing on making inferences within clusters.

Regarding your query about testing for interaction effects, while we recognize the importance of such causal analysis for future research, our current study did not aim to explore potential synergistic or antagonistic effects among covariates on guideline adherence. We plan to conduct future analyses looking further at the role of site and context-specific factors influencing adherence.

We appreciate the opportunity to clarify these aspects and thank you again for your valuable feedback.

Did authors try to quantify any rural-urban, and type of health infrastructure effect in the analyses as these will have potential impact on the adherence level?

Response:

Thank you for your question. As mentioned above, our analysis aimed to provide marginal estimates of the effect of correlates across hospital sites. While we recognize the potential impact of rural-urban classification and the type of health infrastructure on adherence levels as you rightly pointed out, these factors were not separately quantified in the analyses. Instead, they were accounted for by the design—through clustering by hospital site in the Generalized Estimating Equations (GEE) model. This approach allowed us to address the potential variability in adherence due to these factors without directly modeling them as separate covariates.

I think Table 1 needs a better presentation. Number and respective percent value can be presented as 'n (%)' in the same column for each country. The total column should be the first column, and authors should also avoid the vertical separating lines.

Response:

Thank you. Table 1 is now revised as suggested. Number and respective percent value are now presented as 'n (%)' in the same column for each country. The total column is now the first column, and the vertical separating lines are omitted.

Table 2 is quite innovative showing different levels of adherence using different shades of blue colour. However, it does not provide the range/cut-off for each level of adherence either in the title or in the table footnote.

Response:

Thank you for your feedback. We have elaborated upon information previously included in the footnotes and moved this to a subtitle of the table so the information is more accessible to the reader. We have attempted to summarize a large amount of information included in Supplemental Table 4, by visually indicating how the majority of patients were treated for each sub-condition in each site. For example, in Burkina Faso the majority of patients presenting with non-severe pneumonia were treated with fully non adherent care (dark blue cell). In contrast, the majority of patients presenting with non-severe pneumonia in Uganda were treated with fully adherent care (light blue cell). Thus, no cut offs were used. Instead, we are attempting to summarize how the majority (e.g. highest proportion) of patients were treated.

I think Table 3 better suits in the supplementary as Table 2 already provided the summary of the adherence level and its results can be accompanied with the Table 2 description.

Response:

Thank you for your thoughtful suggestion. After careful consideration, we would prefer to leave Table 3 in the body of the paper because it summarizes guideline-specific recommendations. We believe this complements the broader overview of condition-specific adherence presented in Table 2. This visual presentation of recommendation-specific and condition-specific adherence is crucial to our findings and discussion.

Why did authors chose reporting only univariate analysis in the Table 4 although its description suggests multivariable regression analysis? Please check and rectify as appropriate.

Response:

Thank you for bringing this discrepancy to our attention regarding the Table 4 title and its description in the text. We have reviewed and confirmed that Table 4 indeed presents the results of a multivariable analysis as described in the text. Since we opted for a full model approach during the previous review cycle, there was no univariable analysis involved. The current table shows the full model results. We have revised the table title to accurately reflect that it displays the outcomes of a multivariable regression analysis.

Limitation section should also be addressing the issue of residual confounding in the analysis.

Response:

Thank you for your suggestion. We added the following text in the paragraph discussing the study's limitations.

“We also acknowledge the potential influence of unmeasured confounders such as clinical experience and expertise of different healthcare professionals who treated children within study facilities. Future research would benefit from collecting patient-level data to understand the potential association between provider characteristics and patient-level guideline adherence.”

REVIEWER	Hendrickson, Marissa University of Minnesota Masonic Children's Hospital, Pediatrics
REVIEW RETURNED	16-Jan-2024

GENERAL COMMENTS	Thank you for your response, and for making additional changes for clarity. I find the new subtitle of Table 2 provides a clearer explanation of the meaning of the colors, and the new footnote structure appropriately clarifies the choice made when numbers were equal. My only small concern is that I continue to find the new subtitle slightly misleading. You state: “Cell colors indicate which adherence level applied to the largest proportion of patients, by condition and site. Light blue indicates a majority of patients were treated with full adherence, medium blue indicates a majority of patients were treated with partial adherence, and dark blue indicates a majority of patients were treated with non-adherent care.” The first sentence is appropriate, but the use of the word “majority” in the second one implies to me that the number of patients in the category should be $\geq 50\%$. With three options, it is not always the case that the largest number is a majority. Examples of this include pneumonia (overall) and diarrhea (overall), neither of which has any group with $\geq 50\%$ of patients. I would recommend replacing "majority" with "the largest number," which is more accurate. I believe this manuscript is appropriate for publication at this point in either case. Thank you for the opportunity to review it, and for your thoughtful edits.
---

REVIEWER	Bajpai, Ram Keele University, School of Primary, Community, and Social Care
REVIEW RETURNED	18-Jan-2024

GENERAL COMMENTS	Authors have appropriately responded my earlier queries and I have no further comments/questions.
---

VERSION 3 – AUTHOR RESPONSE

Reviewer: 1

Dr. Marissa Hendrickson, University of Minnesota Masonic Children's Hospital

Comments to the Author:

Thank you for your response, and for making additional changes for clarity. I find the new subtitle of Table 2 provides a clearer explanation of the meaning of the colors, and the new footnote structure appropriately clarifies the choice made when numbers were equal.

My only small concern is that I continue to find the new subtitle slightly misleading. You state:

“Cell colors indicate which adherence level applied to the largest proportion of patients, by condition and site. Light blue indicates a majority of patients were treated with full adherence, medium blue

indicates a majority of patients were treated with partial adherence, and dark blue indicates a majority of patients were treated with non-adherent care.”

The first sentence is appropriate, but the use of the word “majority” in the second one implies to me that the number of patients in the category should be $\geq 50\%$. With three options, it is not always the case that the largest number is a majority. Examples of this include pneumonia (overall) and diarrhea (overall), neither of which has any group with $\geq 50\%$ of patients. I would recommend replacing "majority" with "the largest number," which is more accurate.

I believe this manuscript is appropriate for publication at this point in either case. Thank you for the opportunity to review it, and for your thoughtful edits.

Response:

Thank you for your thoughtful comment. We understand your concern regarding the word “majority” in the Table 2 subtitle. We have replaced the “majority” with “largest proportion” in the subtitle.

Reviewer: 3

Dr. Ram Bajpai, Keele University

Comments to the Author:

Authors have appropriately responded my earlier queries and I have no further comments/questions.

Response:

We sincerely thank you for your valuable input to the manuscript thus far.

Reviewer: 1

Competing interests of Reviewer: None

Reviewer: 3

Competing interests of Reviewer